# Buffering and total calcium levels determine the presence of oscillatory regimes in cardiac cells

**Miquel Marchena[1], Blas Echebarria[1], Yohannes Shiferaw[2], Enrique Alvarez-Lacalle[1]**

**1** Departament de Física, Universitat Politècnica de Catalunya-BarcelonaTech, Barcelona, Spain, **2** Physics Department, California State University, Northridge, California 91330, USA

\* enric.alvarez@upc.edu

**Data Availability Statement:** All relevant data are within the manuscript and its Supporting Information files.

## Abstract

Calcium oscillations and waves induce depolarization in cardiac cells which are believed to cause life-threatening arrhythimas. In this work, we study the conditions for the appearance of calcium oscillations in both a detailed subcellular model of calcium dynamics and a minimal model that takes into account just the minimal ingredients of the calcium toolkit. To avoid the effects of homeostatic changes and the interaction with the action potential we consider the somewhat artificial condition of a cell without pacing and with no calcium exchange with the extracellular medium. Both the full subcellular model and the minimal model present the same scenarios depending on the calcium load: two stationary states, one with closed ryanodine receptors (RyR) and most calcium in the cell stored in the sarcoplasmic reticulum (SR), and another, with open RyRs and a depleted SR. In between, calcium oscillations may appear. The robustness of these oscillations is determined by the amount of calsequestrin (CSQ). The lack of this buffer in the SR enhances the appearance of oscillations. The minimal model allows us to relate the stability of the oscillating state to the nullcline structure of the system, and find that its range of existence is bounded by a homoclinic and a Hopf bifurcation, resulting in a sudden transition to the oscillatory regime as the cell calcium load is increased. Adding a small amount of noise to the RyR behavior increases the parameter region where oscillations appear and provides a gradual transition from the resting state to the oscillatory regime, as observed in the subcellular model and experimentally.

## Author summary

In cardiac cells, calcium plays a very important role. An increase in calcium levels is the trigger used by the cell to initiate contraction. Besides, calcium modulates several transmembrane currents, affecting the cell transmembrane potential. Thus, dysregulations in calcium handling have been associated with the appearance of arrhythmias. Often, this dysregulation results in the appearance of periodic calcium waves or global oscillations, providing a pro-arrhythmic substrate. In this paper, we study the onset of calcium oscillations in cardiac cells using both a detailed subcellular model of calcium dynamics and a

**Funding:** M. Marchena, B. Echebarria and E. Alvarez-Lacalle received financial support from: a) Fundació La Marató de TV3, under grant number 20151110, https://www.ccma.cat/tv3/marato/projectes-financats/2014/150/; b) Spanish Ministerio de Economía y Competitividad (MINECO) under grant number SAF2017-88019-C3-2-R https://sede.micinn.gob.es/stfls/eSede/Ficheros/2017/Propuesta_Resolucion_Provisional_Proyectos_Retos_2017.pdf; Y.Shiferaw NHLBI grant R01-HL119095 http://grantome.com/grant/NIH/R01-HL119095-04. The funders had no role in study design, data collection and analysis, decision to publish, or preparation of the manuscript.

**Competing interests:** The authors have declared that no competing interests exist.

minimal model that takes into account the essential ingredients of the calcium toolkit. Both reproduce the main experimental results and link this behavior with the presence of different steady-state solutions and bifurcations that depend on the total amount of calcium in the cell and in the level of buffering present. We expect that this work will help to clarify the conditions under which calcium oscillations appear in cardiac myocytes and, therefore, will represent a step further in the understanding of the origin of cardiac arrhythmias.

## Introduction

Cardiovascular diseases represent one of the main causes of death worldwide [1]. Often, mortality is related to the appearance of rapid cardiac rhythms, such as tachycardia and fibrillation, that result in contractibility loss, reducing cardiac output and eventually leading to sudden cardiac death [2]. Although the onset of rapid arrhythmias can be due to a large variety of factors [3], including changes in the properties of cardiac tissue [4], often arrhythmias are triggered by spontaneous intracellular calcium releases [5, 6]. In cardiac cells, calcium is responsible for regulating cell contraction, but it also modulates several currents that affect the action potential. Thus, spontaneous calcium release in the interbeat interval, during diastole, may elicit extra action potential depolarizations and excitation waves, potentially disrupting normal wave propagation. This sometimes leads to the formation of rotors (functional reentry) and eventually a disordered electrical state characteristic of fibrillation [7–9].

Often, this focal activity is due to the presence of periodic calcium waves, that result in calcium oscillations [10–15]. In paced cardiac cells, oscillations necessarily compete with the external pacing frequency and they may be behind occurrences of spontaneous calcium release events during diastole [16]. Calcium oscillations arise typically due to a malfunction of the Ryanodine Receptor (RyR) [16–19], a ligand-gated calcium channel [20] that controls the amplitude of the intracellular calcium transient, by regulating the release of calcium stored at the sarcoplasmic reticulum (SR). Since calcium dynamics in cardiac cells is regulated by the release of calcium at several tens of thousands of RyR clusters (termed calcium release units, CaRUs), global oscillations must appear as a result of an oscillatory regime at the local cluster level that can later be coordinated by diffusion of free calcium. Alternatively, when synchronization is not complete, oscillations at the local level can give rise to periodic calcium waves, providing a pro-arrhythmic substrate [21, 22]. Calcium oscillations have been observed to appear in ventricular myocytes under elevated values of cytosolic calcium [23], due to periodic opening and closing of the RyRs. An increase in cytosolic calcium concentration results in a higher frequency of the oscillations until, at larger values, the SR is depleted because the RyR becomes permanently open [23]. A similar transition has also been studied in models under conditions of SR calcium overload [24–26].

A key element in the regulation of intracellular calcium dynamics is Calsequestrin (CSQ). This is one of the major $Ca^{2+}$-binding proteins in the SR, with high capacity and low affinity [27]. CSQ is preferentially anchored close to the RyR channels [28, 29]. Nowadays it is known that CSQ is an important regulator of RyR gating [30–32] through the calcium bound to CSQ. Therefore, CSQ plays two roles in cardiac cells: it can be considered a $Ca^{2+}$ reservoir for release from the luminal space, but it is also thought to act as a modulator of RyR channel gating. So that, in addition to simply storing $Ca^{2+}$, CSQ influences the release process by controlling free $Ca^{2+}$ dynamics near the luminal regulatory sites of the RyR complex [33–35]. Dysregulations in CSQ dynamics have been associated with heart diseases [36, 37]. In particular, it has been

shown that the absence of CSQ enhances the RyR channel opening and, thus, promotes premature spontaneous SR Ca$^{2+}$ macro events. The ablation of CSQ causes a form of tachycardia, namely, catecholaminergic polymorphic ventricular tachycardia (CPVT) [36].

In this paper, we use a detailed subcellular calcium model [38] to show the appearance of periodic calcium waves and then analyze this phenomenon using a deterministic model of calcium in a cardiac cell (or in a CaRU). Within this model, we study the existence and stability of different solutions and their dependence on CSQ levels and function. We show that oscillations typically appear at high global calcium concentration and/or high RyR open probability. Their appearance depend on a delicate balance between the total calcium level in the cell and the level of buffering of calcium available. For instance, at high values of CSQ, the system presents a transition from a low concentration, excitable state, to a high concentration state. Such a transition has been proposed to be the basis of complex states, such as long-lasting sparks [39]. At low concentrations of CSQ, in between these two stable states, oscillations appear. We study this transition using a minimal model, that includes the concentration of dyadic and SR calcium and the open probability of the RyR and show that it suffices to explain the appearance of oscillations. A further reduction to a minimal two-dimensional model allows us to explain the transition to the oscillatory regime in terms of the nullcline structure of the system.

## Materials and methods

In this paper we introduce two different approaches to understand Ca$^{2+}$ oscillations in cardiac cells. First, we use a fully detailed subcellular stochastic model of calcium handling to report numerical results showing calcium oscillations. We analyze under which conditions oscillations appear in a controlled scenario where no external pacing is present, and there are no calcium fluxes with the extracellular medium. Later, to gain insight regarding the origin of the oscillations that we observe in the full model, we construct a minimal deterministic model for the local dynamics of calcium at the level of the Calcium Release Unit. The numerical and mathematical analysis of this model allows us to analyze the underlying mechanism of Ca$^{2+}$ oscillations disregarding the coordination effects of the full model.

### Detailed subcellular calcium model

We model the spatial structure of the cell as in a previous model of a cardiomyocyte presented in Marchena and Echebarria [38], which has been modified to add the effects of calsequestrin. A full description of the model is given in the S1 File. We provide now its key features. The equations of the model read:

$$\frac{dc_i(\mathbf{r}, t)}{dt} = J_{rel}(\mathbf{r}, t) - J_{up}(\mathbf{r}, t) + \nabla \cdot [D_i(\mathbf{r})\nabla c_i(\mathbf{r}, t)] - J_{bi}(\mathbf{r}, t) \tag{1}$$

$$\frac{dc_{sr}^{tot}(\mathbf{r}, t)}{dt} = -\frac{v_i(\mathbf{r})}{v_{sr}(\mathbf{r})}[J_{rel}(\mathbf{r}, t) - J_{up}(\mathbf{r}, t)] + \nabla \cdot [D_{sr}(\mathbf{r})\nabla c_{sr}(\mathbf{r}, t)] \tag{2}$$

$$\frac{dc_{bi}(\mathbf{r}, t)}{dt} = J_{bi}(\mathbf{r}, t) \tag{3}$$

where $c_i$ is the calcium concentration in the cytosol, $c_{sr}^{tot}$ the total calcium concentration in the SR, and $c_{bi}$ represents the concentration of a given buffer in the cytosol. Besides, $J_{rel}$ and $J_{up}$ are the release flux from the SR and the uptake by SERCA, respectively, and $J_{bi}$ represents the binding of free calcium to the different buffers in the cytosol (TnC, SR binding buffer and

CaM). These currents are given by:

$$J_{rel} = g_{rel} O_{RyR}(c_{sr} - c_i) \tag{4}$$

$$J_{up} = g_{up} \frac{(c_i/K_i)^2 - (c_{sr}/K_{sr})^2}{1 + (c_i/K_i)^2 + (c_{sr}/K_{sr})^2} \tag{5}$$

$$J_{bi} = k_{on,i} c_i (B_T - c_{b,i}) - k_{off,i} c_{bi}. \tag{6}$$

All the details of the spatial model structure and the values of the parameters can be found in Marchena and Echebarria [38]. The spatial structure of the model includes cytoplasmic and SR spaces, with a spatial discretization of 100 nm. The volume fraction between cytosolic and SR spaces, $v_i/v_{sr}$, is considered to vary spatially, with different values whether the point is close to the z-line or in the inter z-line space. The release flux $J_{rel}$ carries Ca²⁺ ions from the SR to the cytoplasm through the RyRs. In our model, the RyR channels, indicated by a yellow dot in Fig 1a, are distributed over the cell along the z-lines, that we identify with periodic narrow strips (0.3 $\mu$m width) with a predefined period ($T_x$). Experimental data shows that the SR domain coincides with these z-lines [40]. A collection of grid points presenting RyRs forms a cluster, i.e., a CaRU. In cardiac cells, CaRUs are arranged periodically in the longitudinal and transversal directions, with some—seemingly Gaussian—dispersion [41]. We place the centers of the clusters on the perimeter following an exact periodic distribution with a period $T_x = T_y$ = 0.5$\mu$m. Inside the cell, CaRUs are placed following a Gaussian distribution centered at the z-lines and with a fixed dispersion $\sigma = 0.4\mu$m. The average distance between CaRUs is $T_x$ = 1.6$\mu$m and $T_y$ = 0.5$\mu$m. We consider that a CaRU contains 36 RyRs, divided equally among 4 grid points, each one containing 9 RyRs. Each RyR can be in one of four states: open ($O$), close ($C$) and two inactivated states ($I_1$ and $I_2$) as it is shown in Fig 1b. The transitions among these states is considered to be stochastic. In the release flux, the variable $O_{RyR}$ is the fraction of RyRs that are in the open state and is calculated for all grid points that have a group of RyRs.

We have considered two types of RyR regulation. First, we have considered an open rate and termination dependent on both cytosolic and luminal calcium. The dependence on luminal calcium is due to the presence of CSQ, which mediates its opening, as described in the introduction. However, the core of the paper uses a RyR which is not affected by luminal calcium. One of the main goals of this paper is to test which are the effects of buffering on the appearance of oscillations purely in terms of its binding properties to calcium and not its

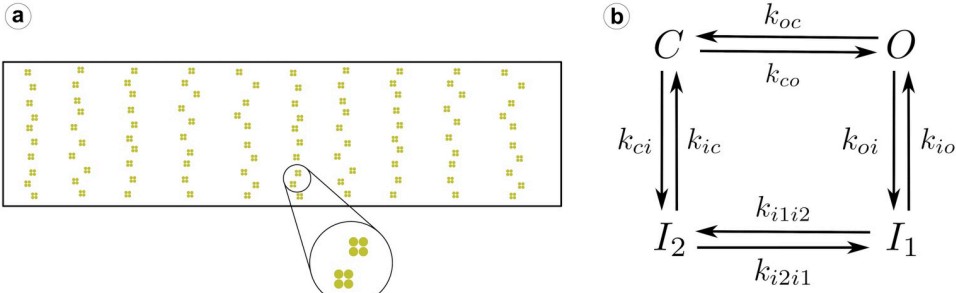

**Fig 1.** a) RyR distribution in the cell. Each CaRU is formed by four simulation voxels, each one containing 9 RyRs. Thus, all CaRUs are formed by 36 RyRs. The CaRUs are distributed over the cell along the z-lines with a Gaussian distribution in both transversal and longitudinal axes. b) Each RyR follows a four state model, with stochastic transitions among the different states.

regulatory effects. Introducing a CSQ-mediated RyR opening would mix different possible mechanisms. For comparison and analysis, we think it is better to leave this dependence initially out. Once the mechanism of CSQ as a buffer is understood we have used section 2 in the S1 File. to show that a full detailed model of RyR with its open rate dependent on SR calcium content gives exactly the same transition to oscillations as described in the main text. If anything, this regulatory mechanism just makes them more robust.

Besides the presence of CSQ in the SR, we also consider in the model the concentrations of other buffers. In particular, TnC, CaM and SR-bound buffers in the cytosol [38]. Due to the addition of CSQ in the model, two parameters have been adjusted from the parameters published in Marchena and Echebarria [38]: the opening rate parameter, now $k_a = 2.1 \cdot 10^{-3}$ $\mu M^{-2} ms^{-1}$, and the dependence of the open probability of the RyR on luminal calcium, now $EC_{50-SR} = 450\ \mu M$. Contrary to the buffers in the cytosol, the dynamics of CSQ is considered to be faster [42–44] than release with a time scale hundreds of time shorter. If we denote by $c_{bSQ}$ the calcium concentration bound to CSQ in the SR, then, the amount of bound calcium is given by:

$$\frac{dc_{bSQ}}{dt} = k_{onSQ}c_{sr}(B_{SQ} - c_{bSQ}) - k_{offSQ}c_{bSQ} \tag{7}$$

Assuming fast binding, the stationary condition for $c_{bSQ}$ ($\dot{c}_{bSQ} = 0$) is:

$$c_{bSQ} = \frac{B_{SQ}c_{sr}}{K_{SQ} + c_{sr}} \tag{8}$$

where $K_{SQ} = k_{offSQ}/k_{onSQ}$ is the dissociation constant. From this, the concentration of free calcium can be obtained solving Eq (2) for the total amount of calcium in the SR, $c_{sr}^{tot}$

$$c_{sr}^{tot} = c_{sr} + c_{bSQ} = c_{sr} + \frac{B_{SQ}c_{sr}}{K_{SQ} + c_{sr}} \tag{9}$$

Solving this equation we obtain the value of free calcium in the SR,

$$c_{sr} = \frac{1}{2}\left[c_{sr}^{tot} - K_{SQ} - B_{SQ} + \sqrt{\left(c_{sr}^{tot} - K_{SQ} - B_{SQ}\right)^2 + 4c_{sr}^{tot}K_{SQ}}\right] \tag{10}$$

The advantage of using this formulation of the rapid buffer approximation over the more usual, for example in [45], is that it conserves mass exactly.

Under physiological conditions, the total amount of calcium in the cell at steady state is fixed by calcium homeostasis, i.e. the complex interaction of LCC, exchanger, and pumps, which affect the steady state level at which the calcium entering the cell balances the calcium extruding. In this work, we are interested in studying instabilities in calcium cycling, under constant cell calcium content. This allows us to focus the analysis on the conditions for the appearance of calcium oscillations under different possible calcium homeostatic levels. Thus, we neglect calcium exchange with the extracellular medium, setting the conductances of the L-type calcium channels and the NCX equal to zero. Then, the total amount of calcium in the cell, $Q_T$, is given by:

$$Q_T = v_i(c_i + c_{b,TnC} + c_{b,SR} + c_{b,CaM}) + v_{sr}(c_{sr} + c_{bSQ}). \tag{11}$$

For a better comparison with the results from a reduced calcium model, described later, we will consider as our control parameter the average calcium content of the cell $\bar{c}_T = Q_T/(v_i + v_{sr})$. Thus, in our simulations, $\bar{c}_T$ is a constant value that is determined by the initial conditions for cytosolic and luminal calcium (free and bound to buffers).

## Reduced calcium model

The minimal model for the local dynamics of calcium is based on the schematics shown in Fig 2. We consider a simplified description of the system, with dynamics of the total calcium concentration in the SR, $c_{SR}^{tot}$, and in the cytosolic space close to the RyR2, or dyadic space, $c_d$, and of the open probability of the RyR, $P_o$,

$$\frac{dc_d}{dt} = J_{rel} - J_d \tag{12}$$

$$\frac{dc_{sr}^{tot}}{dt} = \frac{v_i}{v_{sr}}J_{up} - \frac{v_d}{v_{sr}}J_{rel} \tag{13}$$

$$\frac{dP_o}{dt} = k_p c_d^2 (1 - P_o) - k_m P_o \tag{14}$$

with the currents given by

$$J_{rel} = gP_o(c_{sr} - c_d), \quad J_d = \frac{c_d - c_i}{\tau_i}, \quad J_{up} = g_{up}\frac{c_i^2}{K_s^2 + c_i^2} \tag{15}$$

A detailed derivation of these equations and their range of validity can be found in Section 3 of the S1 File. Notice that, as in the full model, we consider a situation where no external pacing is imposed. In this sense, neither external intake from the LCC is considered, nor any extrusion via the sodium-calcium exchanger.

For simplicity, we consider a SERCA pump without an equilibrium condition, that always pumps calcium from the cytosol to the SR (see Eq (15). This gives a trivial solution at $c_i = c_d = 0$, instead of the physiological value of $\sim 100\text{nM}$. However, at these values of $c_d$ and $c_i$ the luminal calcium is roughly 1mM and, thus, the trivial solution is a reasonable simplification. As in the detailed subcellular model, we assume the approximation of rapid CSQ buffer, so we can compute the amount of free luminal calcium $c_{sr}$ from the total luminal calcium $c_{sr}^{tot}$ from Eq (9).

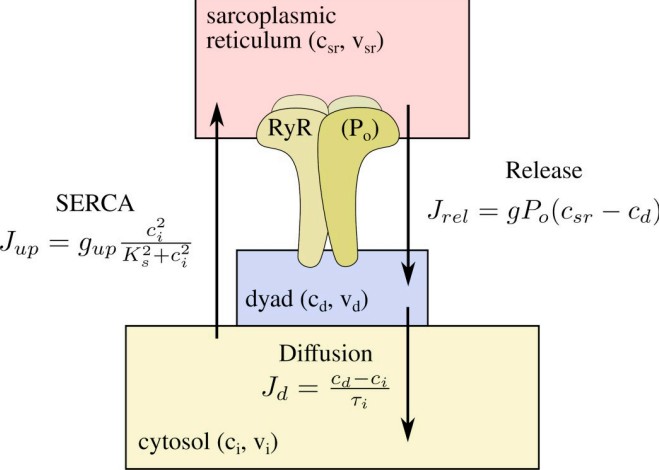

**Fig 2. Sketch of the different compartments considered in the simplified model, with the internal variables and the equations of the respective calcium fluxes.**

To close the system we should introduce an extra equation for calcium concentration in the cytosol $c_i$. However, as we assume that the total calcium content in the cell is constant, then we have a conservation equation. Therefore, we can compute $c_i$ solving the following quadratic equation for the conservation of $\bar{c}_T$

$$\bar{c}_T = \frac{v_i}{v}\left(c_i + \frac{B_b c_i}{K_b + c_i}\right) + \frac{v_d}{v}c_d + \frac{v_{sr}}{v}c_{sr}^{tot}, \tag{16}$$

where $v$ is the unit volume defined as $v \equiv v_i + v_{sr} + v_d$, and $B_b$ is the concentration of a generic buffer in the cytosol.

To simplify the analysis, we proceed to work with the assumption that the dynamics of opening of the RyRs is faster than that of calcium concentration ($\dot{P}_o \simeq 0$), obtaining then a minimal two-variable model. This will be our baseline minimal model. However, we will later also consider an alternative model with fast dynamics for the dyadic calcium concentration ($\dot{c}_d \simeq 0$). Clearly, under the assumption that both processes are fast, as considered often in the literature [46], in the model oscillations would dissappear.

A third theoretical possibility would be to consider fast luminal dynamics. Although analytically possible, this approach does not have much physiological sense, as SERCA is typically slow compared to release or diffusion from the dyadic space.

**Fast RyR dynamics.** In this case, we assume that the open and close dynamics of the RyR are fast, so we can assume that it is in a quasi-steady state ($\dot{P}_o \simeq 0$). Then, from Eq (14), we obtain:

$$P_o = \frac{c_d^2}{K_o^2 + c_d^2} \tag{17}$$

where the parameter $K_o^2 = k_m/k_p$ is the ratio of the open and close rates of the RyR.

Then, with these assumptions, the simplified model becomes

$$\frac{dc_d}{dt} = g\frac{c_d^2}{K_o^2 + c_d^2}(c_{sr} - c_d) - \frac{c_d - c_i}{\tau_i} \tag{18}$$

$$\frac{dc_{sr}^{tot}}{dt} = \frac{v_i}{v_{sr}}g_{up}\frac{c_i^2}{K_s^2 + c_i^2} - \frac{v_d}{v_{sr}}g\frac{c_d^2}{K_o^2 + c_d^2}(c_{sr} - c_d) \tag{19}$$

**Fast dyadic calcium dynamics.** In order to test the robustness of the analysis, we also consider a simplified model given by Eqs (12)–(14), in the limit of fast dynamics in the dyadic space and take $\dot{c}_d \simeq 0$. Then, from Eq (12):

$$c_d = \frac{\tau_i g P_o c_{sr} + c_i}{1 + \tau_i g P_o} \tag{20}$$

Substituting this expression in Eqs (13) and (14), we obtain another minimal model, given by

$$\frac{dc_{sr}^{tot}}{dt} = \frac{v_i}{v_{sr}}g_{up}\frac{c_i^2}{K_s^2 + c_i^2} - \frac{v_d}{v_{sr}}g P_o(c_{sr} - c_d) \tag{21}$$

$$\frac{dP_o}{dt} = -k_m P_o + k_p c_d^2(1 - P_o) \tag{22}$$

where again, $c_i$ must be computed solving the quadratic equation for the conservation of mass $\bar{c}_T$ [Eq (16)]. For simplicity we will consider the case when no calsequestrin is present $B_{SQ} = 0$.

## Results

We first present the results of the numerical simulations of both the full detailed model and the minimal model of calcium cycling. Both produce the same basic scenarios for intracellular calcium dynamics, with three different dynamical behaviors, which we then proceed to analyze. The goal of the development of the minimal model is, precisely, to be able to perform this analytical treatment and check how the behavior depends on total calcium and buffering levels.

### Subcellular model

The full detailed model allows us to investigate the different behaviors present in cardiomyocyte calcium cycling when there is no external pacing. We should point out that, under these conditions, the average calcium concentration in the cell $\bar{c}_T$ is conserved since the total amount of calcium $Q_T$ in the cell is constant. We produce simulations with different levels of average calcium concentration and observe very different behaviors (Fig 3a). For the lowest value of $\bar{c}_T$, the RyR remains almost closed, and most of the calcium content is stored in the SR. Despite the stochasticity of the system, the average values obtained are reproduced reliably with only the presence of local sparks as fluctuations of this global state. This state corresponds to an excitable state, which is the expected behavior of the cell if it has to react properly to external excitation. We call this general state a global shutdown state.

When the calcium load increases, the system starts to spontaneously show calcium waves. These waves persist in time with different shapes and durations, giving rise to a nearly periodic oscillation in the global calcium signal. Roughly, we observe one calcium wave per second (Fig 4). Waves are normally initiated at different sites each time but they appear systematically indicating a strong oscillation at the substrate level that we will address in the discussion.

Finally, at large values of $\bar{c}_T$ oscillations disappear, giving place to a stable state with a high level of calcium in the cytosolic space and a depleted SR. In this state, the RyRs are generally open allowing for the depletion of the SR and the increase of cytosolic calcium. Except for local fluctuations this state is globally stable and we can call it the open state. This state would not respond to external pacing. However, it would produce the activation of the NCX exchanger, which would slowly decrease the average concentration model. As we pointed out previously, the elimination of the calcium intake and extrusion in the model allows us to focus on the general behavior of the cell under different homeostatic scenarios. Numerical simulations indicate that as the calcium level is increased, the cell goes from a shut-down and ready-to-respond state to an oscillatory regime to a global open state where the cell does not respond. Including RyR regulation by SR Ca does not change this scenario, but rather, it actually increases the region where oscillations appear (Fig. 2 in S1 File).

A similar trend has been observed experimentally by Stevens et al [23], even if in the experimental preparation the control parameter was the amount of cytosolic calcium, and not total calcium, as in our simulations. Oscillations appear as the amount of calcium in the cell increases, giving rise to a state with depleted SR calcium (and RyRs in the open state), at high calcium concentrations. Furthermore, experimentally it has been shown that changes in buffering levels can have also important effects on this transition. More specifically, Stevens et. al [23] have shown that the reduction of CSQ in the SR bulk enhances the appearance of oscillations. We have checked if this situation is also present in our simulation and found this to be the case. As shown in Fig 4, when we reduce the CSQ concentration, the oscillations appear at

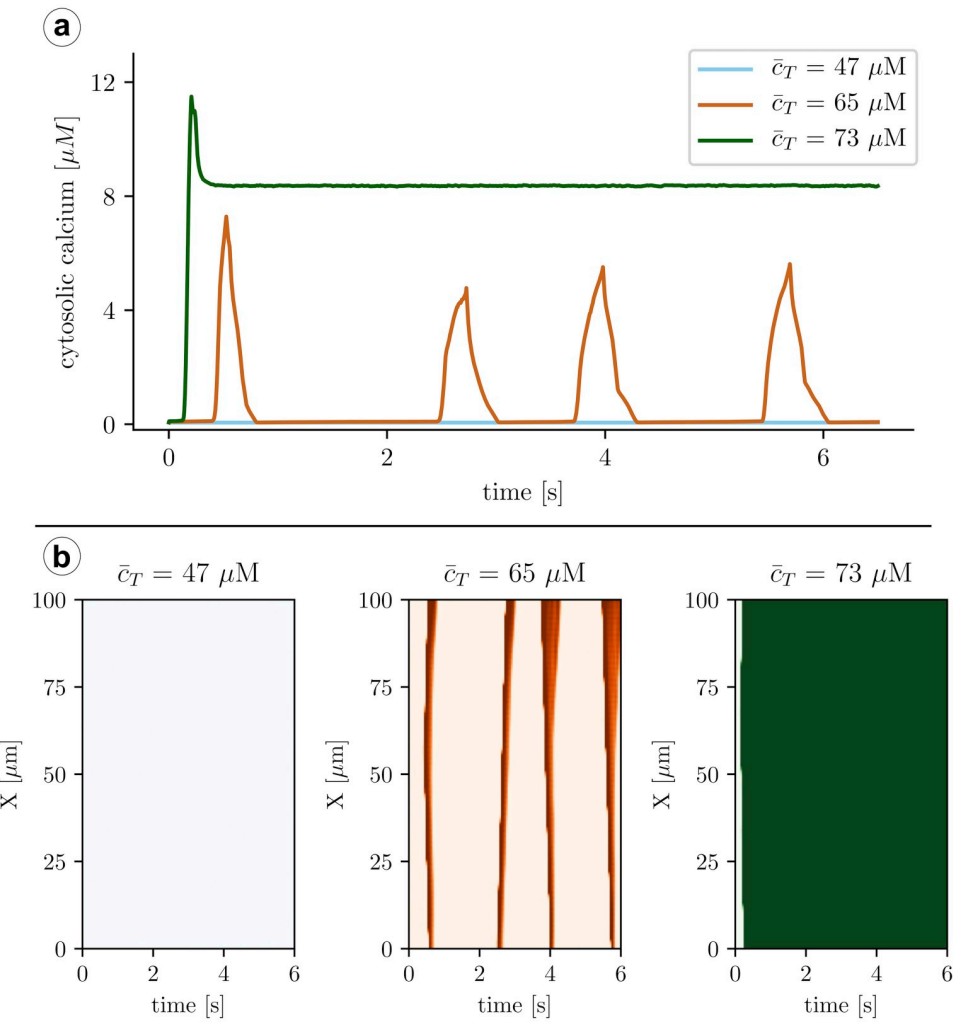

**Fig 3.** A: Calcium traces obtained with the full subcellular model and three different values of the average calcium concentration, $\bar{c}_T$. B: Line-scans at different values of the load. Increasing the load, the system undergoes a transition from a low cytosolic calcium state (at $\bar{c}_T = 47\mu M$), where RyRs remain in the closed state, to spontaneous oscillations, giving rise to calcium waves ($\bar{c}_T = 65\mu M$). Finally, at high calcium loads ($\bar{c}_T = 73\mu M$) oscillations give rise to a high cytosolic calcium state, where the RyRs remain open, resulting in SR calcium depletion.

lower values of $\bar{c}_T$, they have a higher frequency and the range of oscillations in terms of $\bar{c}_T$ becomes broader.

## Minimal model without calsequestrin

We proceed to explain the results obtained in the minimal model where we find the same qualitative behavior as in the results obtained with the full subcellular stochastic model. In this respect, the minimal three variable model, Eqs (12)–(14), reproduces the appearance of oscillations (S1 Fig). However, since we are interested in using these models to obtain a qualitative understanding of the instability, we have rather considered the reduction to two variables. We have performed simulations in the approximation of fast RyR open dynamics at different values of the cell average calcium concentration $\bar{c}_T$. We consider first the case when no calsequestrin is present, $B_{SQ} = 0$. As we observed in the full subcellular model, at low values of $\bar{c}_T$ the

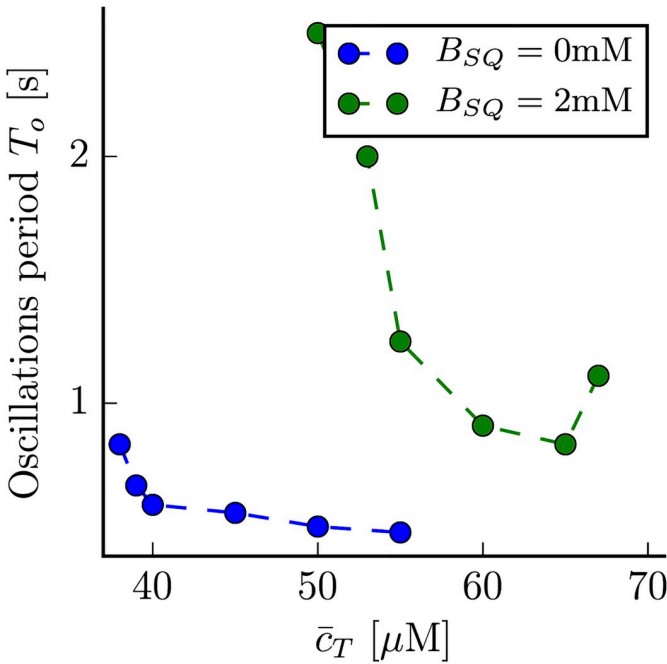

**Fig 4. The average period of oscillations at different values of the average calcium concentration $\bar{c}_T$, for a concentration of CSQ of $B_{SQ}$ = 2mM (green dots), and in the absence of CSQ (blue dots).**

system remains in a low concentration steady state (Fig 5). In this state the system is excitable, so the fixed point is locally stable, but a large enough perturbation produces an increase in calcium concentration, resulting in a calcium transient typical from CICR. On the other hand, at high calcium loads, there is a new fixed point with high cytosolic calcium concentration, that has been related to the appearance of long-lasting sparks [39]. As in the subcellular model, in between the low concentration fixed point and the permanently open state, the system presents oscillations (Fig 5), that are stable for a quite broad range of loads.

Indeed, the number of stationary solutions changes with the calcium concentration $\bar{c}_T$. The fixed points of the system can be found from:

$$0 = g \frac{c_d^2}{K_o^2 + c_d^2} (c_{sr} - c_d) - \frac{c_d - c_i}{\tau_i} \tag{23}$$

$$0 = -\frac{v_d}{v_{sr}} g \frac{c_d^2}{K_o^2 + c_d^2} (c_{sr} - c_d) + \frac{v_i}{v_{sr}} g_{up} \frac{c_i^2}{K_s^2 + c_i^2} \tag{24}$$

together with:

$$c_i + \frac{B_b c_i}{K_b + c_i} = \frac{1}{v_i} \left( v\bar{c}_T - v_d c_d - v_{sr} c_{sr}^{tot} \right) \tag{25}$$

Eqs (23)–(25) represent three algebraic equations that give the concentrations $c_i$, $c_d$ and $c_{sr}$ as a function of total calcium concentration in the cell $\bar{c}_T$. When no calsequestrin is present,

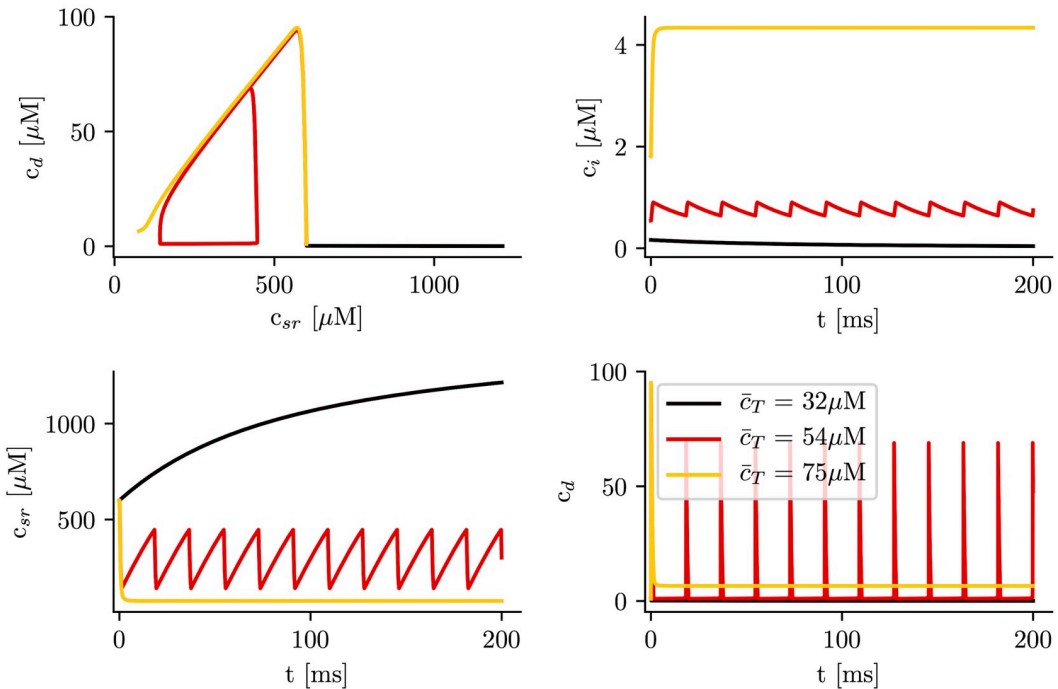

**Fig 5. Time traces of the different calcium concentrations for different values of total calcium concentration, $\bar{c}_T$, and calsequestrin concentration set to zero ($B_{SQ} = 0$).** After a transient, the system ends up in either a steady state which is excitatory at low levels of total calcium in the cell with observed low levels of calcium in the cytosol, in an oscillatory state with intermediate levels of total calcium in the cell, or in a state of high total levels of calcium in the cell with observed high cytosolic calcium levels.

$B_{SQ} = 0$, it is easy to obtain that

$$c_d = c_i + \frac{v_i}{v_d}\tau_i g_{up}\frac{c_i^2}{K_s^2 + c_i^2} \tag{26}$$

$$c_{sr}^{tot} = \frac{1}{v_{sr}}\left[v\bar{c}_T - v_d c_d - v_i\left(c_i + \frac{B_b c_i}{K_b + c_i}\right)\right] \tag{27}$$

Introducing these expressions into Eq (23) we obtain an equation of the form $f(c_i; \bar{c}_T) = 0$. For each value of $\bar{c}_T$ we can obtain the values of $c_i$ that solve the equation. For instance, for a global average calcium concentration of $\bar{c}_T = 32$ $\mu$M we have only one solution, as shown in Fig 6a. This solution, given by $c_d = c_i = 0$ and $c_{sr}^{tot} = v/v_{sr}\bar{c}_T$, exists for all values of $\bar{c}_T$. At high values of the average concentration, $\bar{c}_T = 54$ and $75$ $\mu$M, another two solutions appear, as depicted in Fig 6b and 6c.

To calculate the stability of the stationary solutions, we have computed the value of the eigenvalues of the Jacobian matrix, corresponding to Eqs (18) and (19). We find that, while the lower branch is always stable, the other branch of solutions is unstable for a large range of parameters (Fig 7), due to the appearance of oscillations. The stability of the corresponding periodic orbit has been calculated using XPPAUT [47] (Fig 8). We obtain that, as $\bar{c}_T$ is increased, a limit cycle appears in a global homoclinic bifurcation, with zero frequency (Fig 8c). Increasing $\bar{c}_T$, this limit cycle finally disappears in a Hopf bifurcation, at which the upper fixed point becomes stable (Fig 8b).

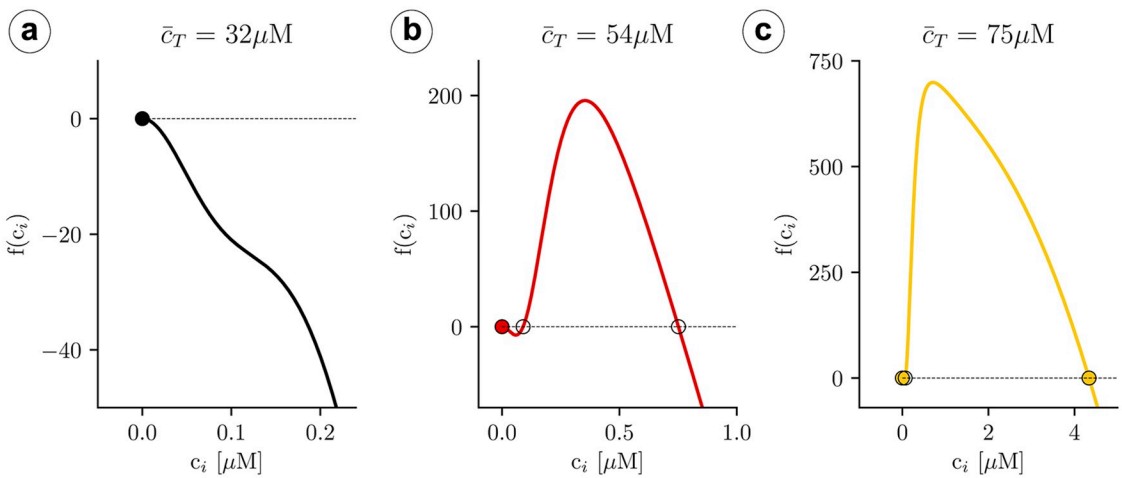

**Fig 6. Plot of the function $f(c_i)$ for different values of the $\bar{c}_T$ concentration.** a) At low concentrations there is a single fixed point.
b) At higher concentrations two extra unstable fixed points appear. c) At high concentrations the upper fixed point becomes stable.

## Minimal model with calsequestrin

We present now the main results of the simulation in the fast RyR minimal model when calse-
questrin is present. We use Eqs (18) and (19), with Eq (9) that relates free with CSQ-bound SR
calcium concentrations. Typically $K_{SQ} = 650\mu M$ and $B_{SQ}$ somewhere between zero and 20
mM. We have then calculated the different fixed points as a function of the total concentration
$\bar{c}_T$ for different values of $B_{SQ}$ (Fig 9). When $B_{SQ} \neq 0$, increasing $\bar{c}_T$, the appearance of two extra
solutions occurs at larger values of $\bar{c}_T$, meaning that the close state solution is stable for a wider
range of $\bar{c}_T$. Besides, the oscillatory range becomes narrower when $B_{SQ}$ increases, until at

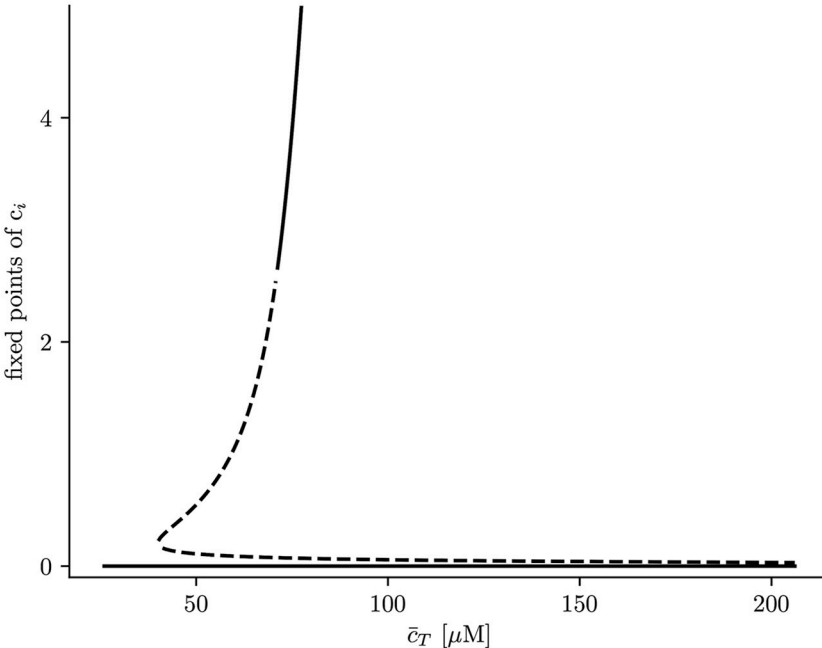

**Fig 7. Solutions for cytosolic calcium concentration, $c_i$, as a function of total calcium concentration, $\bar{c}_T$.**
Discontinuous lines represent unstable solutions while continuous lines stable ones.

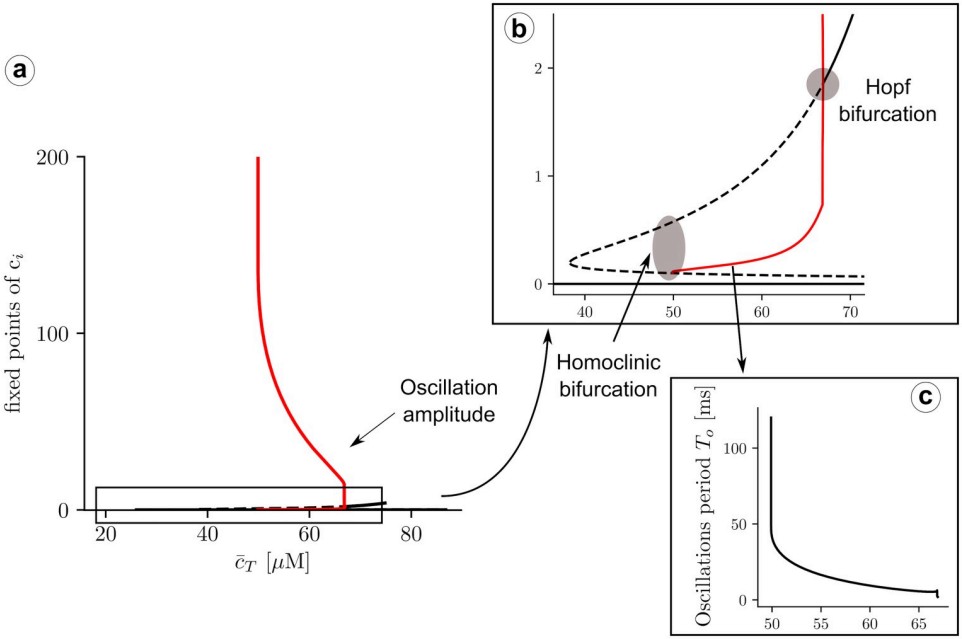

**Fig 8.** a) Solutions for cytosolic calcium concentration, $c_i$, as a function of total calcium concentration, $\bar{c}_T$. Discontinuous lines represent unstable solutions and continuous lines stable ones. A closer look at the transitions is shown in b). When reducing the total concentration, at $\bar{c}_T \approx 66$ $\mu$M, a limit cycle emerges in a Hopf bifurcation, from the upper state, that then becomes unstable. The red lines represent the lower and upper values of the limit cycle. At, $\bar{c}_T \approx 50$ $\mu$M, the intermediate unstable fixed point collides with the limit cycle, that disappears in a homoclinic bifurcation. Below $\bar{c}_T \approx 39$ $\mu$M, the RyR close state is the only solution. c) Oscillation periods as a function of $\bar{c}_T$.

certain point oscillations disappear. The disappearance of oscillations is due to the transformation of the bifurcation at which the upper fixed value gains stability, from a Hopf bifurcation into a saddle-node bifurcation.

In addition, at this point, the system presents five fixed points where only the lowest and uppermost are stable. It is important to note that this shows that CSQ enhances the elimination of oscillations. Finally, for high values of $\bar{c}_T$, the system has again three fixed points.

## Analytical results

The mathematical tractability of the minimal model allows us to get a better understanding of the transition to the upper state via oscillations. A first insight can be obtained by plotting the nullclines of the system (Fig 10), which can help us understand the main mechanisms behind the transition to the oscillatory state. In particular, we obtain the critical average calcium concentration for the onset of oscillations, which depends on buffering levels, and the conditions for the appearance of the upper state.

**Nullclines and stability of solutions.** Besides the transition from one to three solutions (Fig 10a and 10b), nullclines present a clear restructuring of their branches well before the upper state becomes stable. Increasing the total concentration, there is a sudden pinch-off in the $c_d$-nullcline (Fig 10b and 10c). Before this change in nullcline topology, the lower state (with all the calcium in the luminal space and none in the cytosol) is the only stable attractor. Once the $c_d$-nullclines split, oscillations may appear around the upper unstable fixed point.

We can understand the effect of the pinch-off, plotting the trajectories for values of $\bar{c}_T$ close to the transition. Below the pinch-off, the trajectory follows the fast dynamics of the $c_d$-nullcline (the black line in Figs 10 and 11), until it reaches the fixed point. As the load is increased,

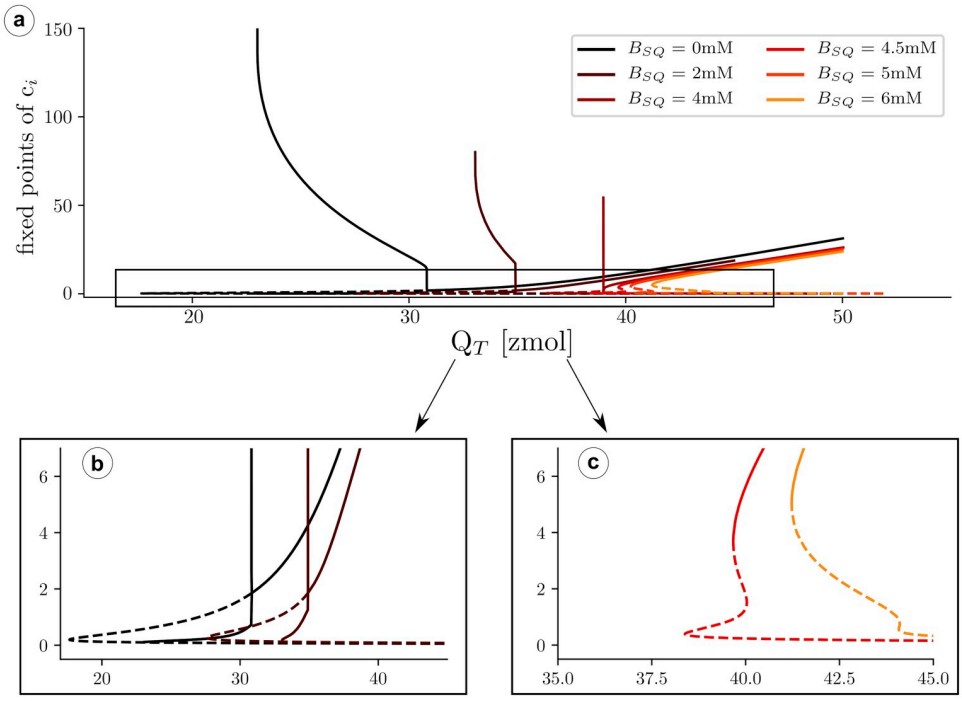

**Fig 9.** a) Number of fixed points and stability of those fixed points as a function of total calcium in the unit, for different values of calsequestrin concentration. In b) and c) details of the transition are shown, for selected values of $B_{SQ}$.

the branches of the $c_d$-nullclines come closer until at a certain point the break-up occurs (Fig 11c). Due to the emergence of the pinch-off, the system dynamics follows the lower nullcline up to the tip, at the largest value of $c_{sr}$, where, due to the fast dynamics in the $c_d$ direction, it jumps to follow again the nullcline, at larger values of $c_i$. Since there is no stable point, the trajectory starts a persistent oscillation around the unstable fixed point. We can state that, for this problem, the nullcline break-up is the necessary and sufficient condition to obtain oscillations.

**Onset of oscillations.** Using this observation, we can calculate analytically the critical value of $\bar{c}_T$ beyond which the system oscillates. At the pinch-off of the nullclines, the cubic solution of $\dot{c}_d = 0$ (black nullcline) loses two of the three solutions at a given $c_{sr}$. To calculate this point analytically, first we observe that the pinch-off occurs at values of $c_d < K_o$ ($K_o = 15\mu$M). To simplify the calculations, let us make the approximations that, at the pinch off, the $c_d$ satisfies $c_d \ll K_o$ and $c_d \ll c_{sr}$. Being this the case, then Eq (18) reduces to

$$\frac{dc_d}{dt} = 0 = g\frac{c_d^2}{K_o^2}c_{sr} - \frac{c_d - c_i}{\tau_i} \tag{28}$$

Furthermore, from Eq (25), we can write $c_i$ in terms of $c_{sr}$ (assuming $B_{SQ} = 0$, $c_d \ll c_{sr}$)

$$c_i + \frac{B_b c_i}{K_b + c_i} = \frac{v\bar{c}_T - v_{sr}c_{sr}}{v_i} \tag{29}$$

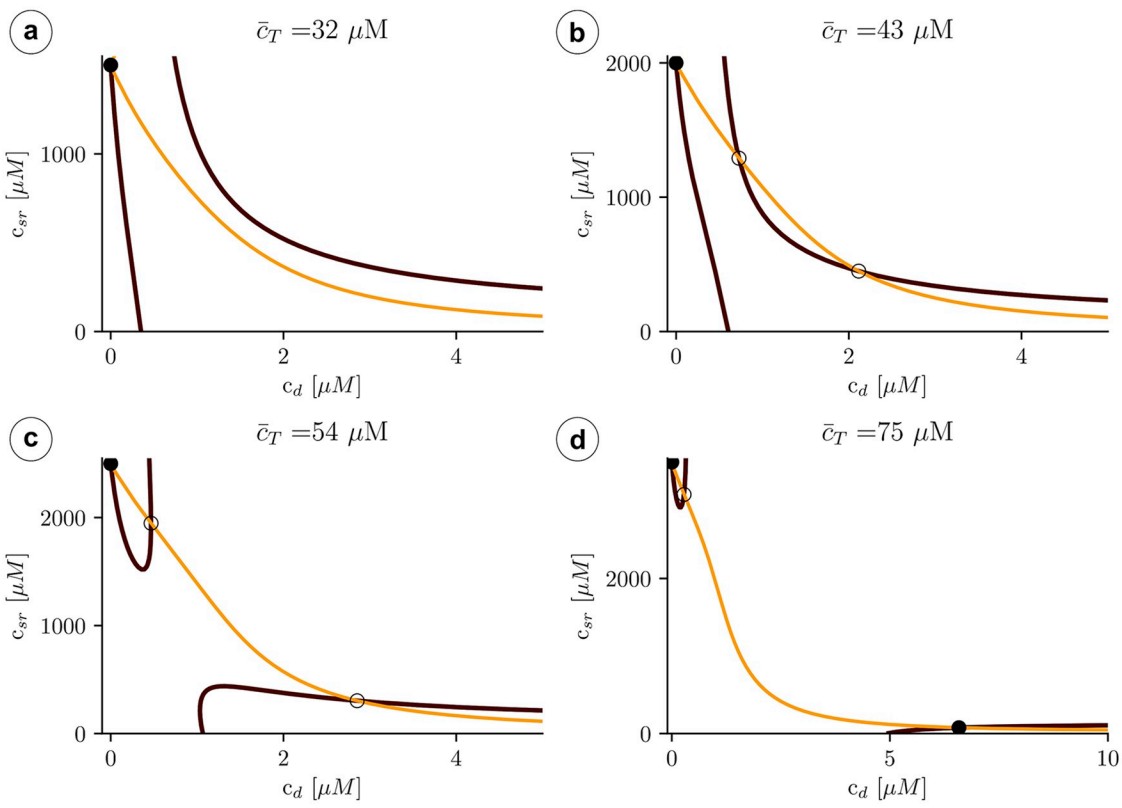

**Fig 10. Structure of the nullclines at different values of $\bar{c}_T$ indicated in the title of each panel.** The black line indicates the first nullcline $\dot{c}_d = 0$, while the orange lines corresponds to $\dot{c}_{sr} = 0$. Dots indicate the fixed points. Filled dot: stable fixed point and unfilled dot: unstable fixed point.

Assuming that the concentration of bound cytosolic calcium is much larger than that of free cytosolic calcium, we obtain

$$c_i = \frac{K_b(v\bar{c}_T - v_{sr}c_{sr})}{v_iB_b - (v\bar{c}_T - v_{sr}c_{sr})} \tag{30}$$

From the polynomial equation for $c_d$, Eq (28), solutions of $c_d$ are lost at values of $c_{sr}$ given by $1 - 4\tilde{g}c_ic_{sr} \leq 0$, with $\tilde{g} = g\tau_i/K_o^2$. Expanding, this gives the critical value of $c_{sr}^*$ as:

$$v_iB_b - v\bar{c}_T + v_{sr}c_{sr}^* - 4\tilde{g}c_{sr}^*K_b(v\bar{c}_T - v_{sr}c_{sr}^*) = 0 \tag{31}$$

The same equation can be written as

$$(c_{sr}^*)^2 + c_{sr}^*\frac{v_{sr} - 4\tilde{g}K_bv\bar{c}_T}{4\tilde{g}K_bv_{sr}} + \frac{v_iB_b - v\bar{c}_T}{4\tilde{g}K_bv_{sr}} = 0 \tag{32}$$

Once the pinch-off has been produced (Fig 10c), there are two values of $c_{sr}^*$, corresponding to the lowest and highest values of the upper and lower nullclines, respectively. Just at the pinch-off these two points merge. This allows to establish the critical value at which the oscillation appears as the one that makes zero the discriminant of the second order polynomial of $c_{sr}^*$. After some algebra, the condition for the critical $\bar{c}_T^*$ becomes

$$(v_{sr} - 4\tilde{g}K_bv\bar{c}_T^*)^2 - 16K_b\tilde{g}v_{sr}(v_iB_b - v\bar{c}_T^*) = 0 \tag{33}$$

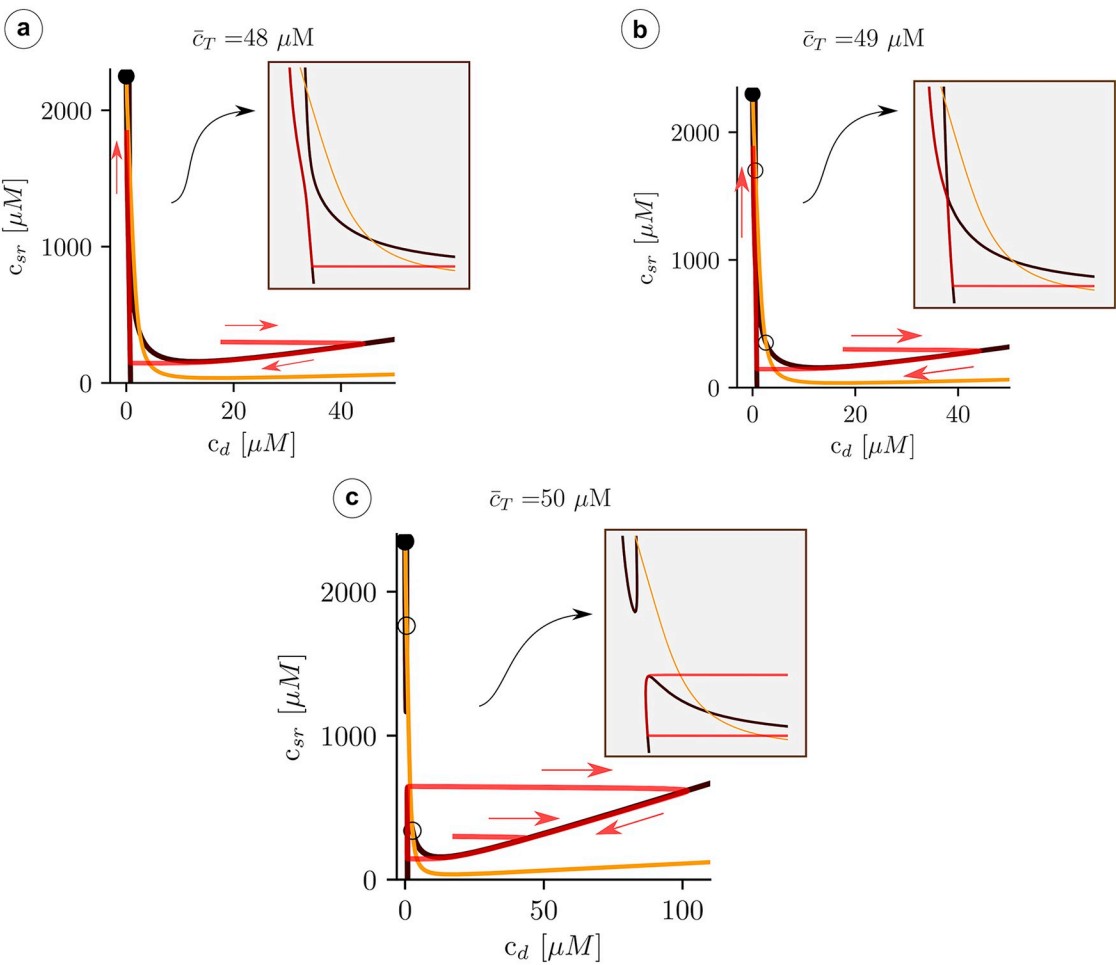

**Fig 11. Structure of the nullclines at different values of $\bar{c}_T$ indicated in the title of each panel.** The black line indicates the nullcline $\dot{c}_d = 0$, while the orange line corresponds to $\dot{c}_{sr} = 0$. The red curve is a trajectory with a direction indicated by the red arrows.

which can be written as

$$\left(\bar{c}_T^*\right)^2 + \frac{v_{sr}}{2\tilde{g}K_b v}\bar{c}_T^* + \frac{v_{sr}^2}{16\tilde{g}^2 K_b^2 v^2} - \frac{v_{sr} v_i B_b}{\tilde{g}K_b v^2} = 0 \tag{34}$$

This gives

$$\bar{c}_T^* = \sqrt{\frac{v_{sr} v_i B_b}{v^2 \tilde{g}K_b} - \frac{v_{sr}}{4\tilde{g}K_b v}} \tag{35}$$

Using the parameters in Table 2 in S1 File, this expression gives a critical value for the onset of oscillations at around $\bar{c}_T^* = 50$ $\mu$M, that given the approximations considered, agrees quite well with the value obtain from the simulations of $\bar{c}_T^* \approx 51$ $\mu$M. Thus, when the total calcium content exceeds this critical value $\bar{c}_T^*$, corresponding to a calcium concentration in the SR of $c_{sr} = v/v_{sr}\bar{c}_T^* = 2.28$mM (in the lower state), the system starts to oscillate, at a value of diastolic

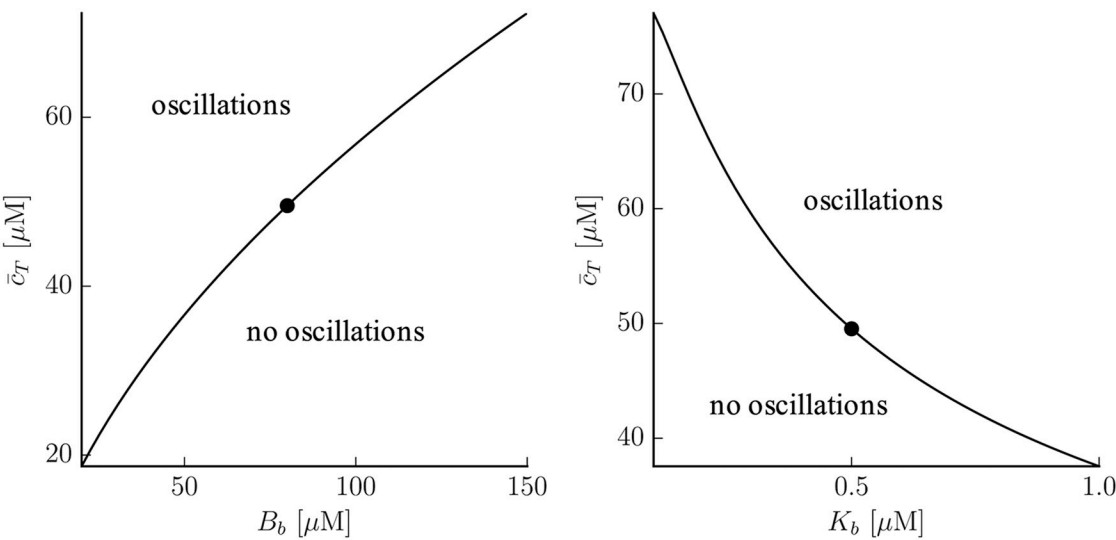

**Fig 12. Dependency of the onset of oscillations with buffer parameters.** The filled dots represent the control values $B_b = 80\mu M$, $K_b = 0.5\mu M$, given in Table 2 in S1 File.

SR calcium load, given by:

$$c_{sr}^* = \frac{v\bar{c}_T^*}{2v_{sr}} - \frac{1}{8\tilde{g}K_b} = \frac{1}{2}\sqrt{\frac{v_i B_b}{v_{sr}\tilde{g}K_b}} - \frac{1}{4\tilde{g}K_b} \tag{36}$$

which gives a value of $c_{sr}^* = 0.86$ mM. At the onset of oscillations, there is thus a sudden decrease in basal SR calcium concentration, to less than half its previous value before the oscillations.

An increase in the quantity of cytosolic buffers (higher $B_b$) results in a delay in the onset of oscillations, that would occur at higher calcium load (Fig 12a). A higher calcium affinity (lower $K_b$) on the contrary, would result in oscillations at lower loads (Fig 12b). It is interesting to notice, too, that the strength of SERCA does not affect the onset of oscillations.

**Transition to the upper state.** The oscillations disappear at a Hopf bifurcation when the upper state becomes stable. It is possible to relate this transition to the structure of the null-clines in Fig 10. For that, let us recover the definition of the Jacobian matrix **J**:

$$\mathbf{J} = \begin{bmatrix} \dfrac{\partial \dot{c}_d}{\partial c_d} & \dfrac{\partial \dot{c}_d}{\partial c_{sr}} \\ \dfrac{\partial \dot{c}_{sr}}{\partial c_d} & \dfrac{\partial \dot{c}_{sr}}{\partial c_{sr}} \end{bmatrix} \equiv \begin{bmatrix} f_1 & f_2 \\ g_1 & g_2 \end{bmatrix} \tag{37}$$

A fixed point will be stable provided $f_1 + g_2 < 0$. When $f_1 g_2 - f_2 g_1 > 0$ the stable fixed point corresponds to a node and if $f_1 g_2 - f_2 g_1 < 0$ to a stable spiral. We can use this to relate the slope of the nullclines to the stability of the upper fixed point. Let us denote $\alpha \equiv \dot{c}_d$ and $\beta \equiv \dot{c}_{sr}$ the time derivatives of the independent variables.

At large values of the total concentration (see Fig 13a, with $\bar{c}_T = 75\mu M$), the slope of the black nullcline ($\alpha = 0$) at the fixed point is positive, while the slope of the orange nullcline ($\beta = 0$) is negative. Then, increasing $c_d$ at constant $c_{sr}$, $\alpha$ goes from being positive to negative. This means that $f_1 \equiv \partial\alpha/\partial c_d < 0$. Using the same argument, it is easy to check that also $g_2 \equiv \partial\beta/\partial c_{sr}$

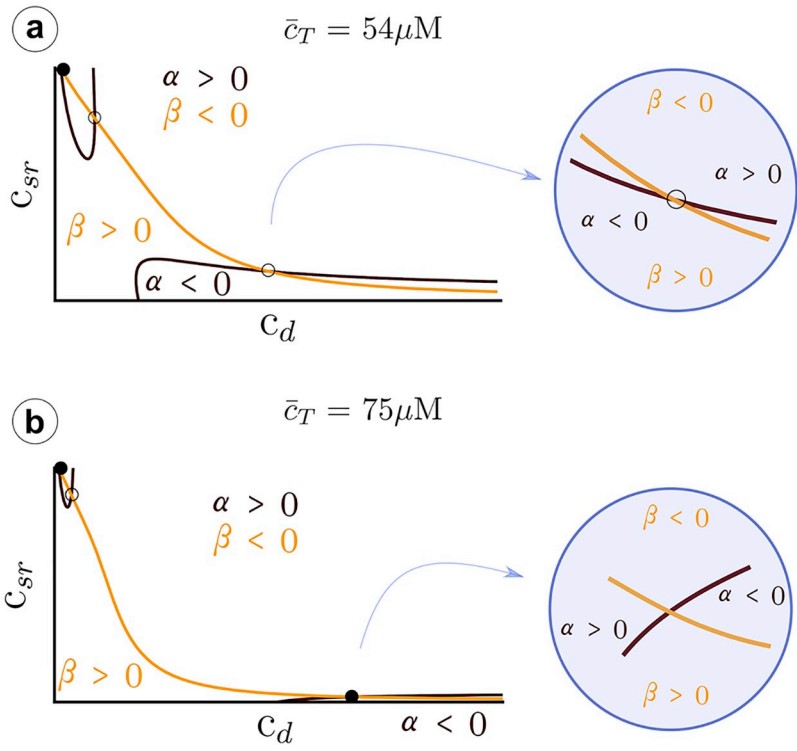

**Fig 13. Structure of the nullclines at two different values of a)** $\bar{c}_T = 54\mu$**M, b)** $\bar{c}_T = 75\mu$**M.** A black line corresponds to the nullcline $\dot{c}_d = 0$, while the orange line indicates $\dot{c}_{sr} = 0$. The functions $f_1$ and $g_2$ are the elements of the diagonal of the Jacobian matrix defined as $f_1 = \partial_{c_d}\dot{c}_d$ and $g_2 = \partial_{c_{sr}}\dot{c}_{sr}$.

$< 0$, and therefore the fixed point is stable. At lower values of the total concentration (see, for instance, Fig 13a, for $\bar{c}_T = 54\mu$M) the slope of the black nullcline becomes negative. Thus, in this case, while $g_2$ is still negative, $f_1$ becomes positive, and it is not possible to determine the stability of the fixed point. It will depend on the speed of rate of $c_d$ and $c_{sr}$ close to the fixed point. If the dynamics of $c_d$ is faster, then one expects this point to be unstable, if it is $c_{sr}$ that varies fast, then stable. One would then expect that buffers that change the dynamics either in the cytosol or in the SR would effect the stability of the fixed point and, therefore, the range of existence of the limit cycle.

## Robustness of the results

**Fast dyadic calcium dynamics.** It is useful to check that the basic points of our discussion hold when different possible approximations are applied to the minimal model. Namely, if the time scale of RyR opening is not as fast as the time scale of calcium diffusion near the dyadic space we should analyze the fast dyadic calcium approximation and not the fast RyR opening approximation to obtain information from the nullcline analysis. To this end, we have performed simulations of Eqs (21) and (22) at different values of the total calcium concentration $\bar{c}_T$ (Fig 14) and test that we find the same basic behavior: at low values of $\bar{c}_T$ the system remains in a low concentration steady state (Fig 15), but oscillations appear for a range of $\bar{c}_T$, up to a limit where an upper state becomes stable.

More importantly, the basic structure of the nullclines determines the possible solutions and again, oscillations appear when pinch off is produced (Fig 14). The fixed points in this case are the same as in Fig 7, since they correspond to fixed points of Eqs (12)–(14). However,

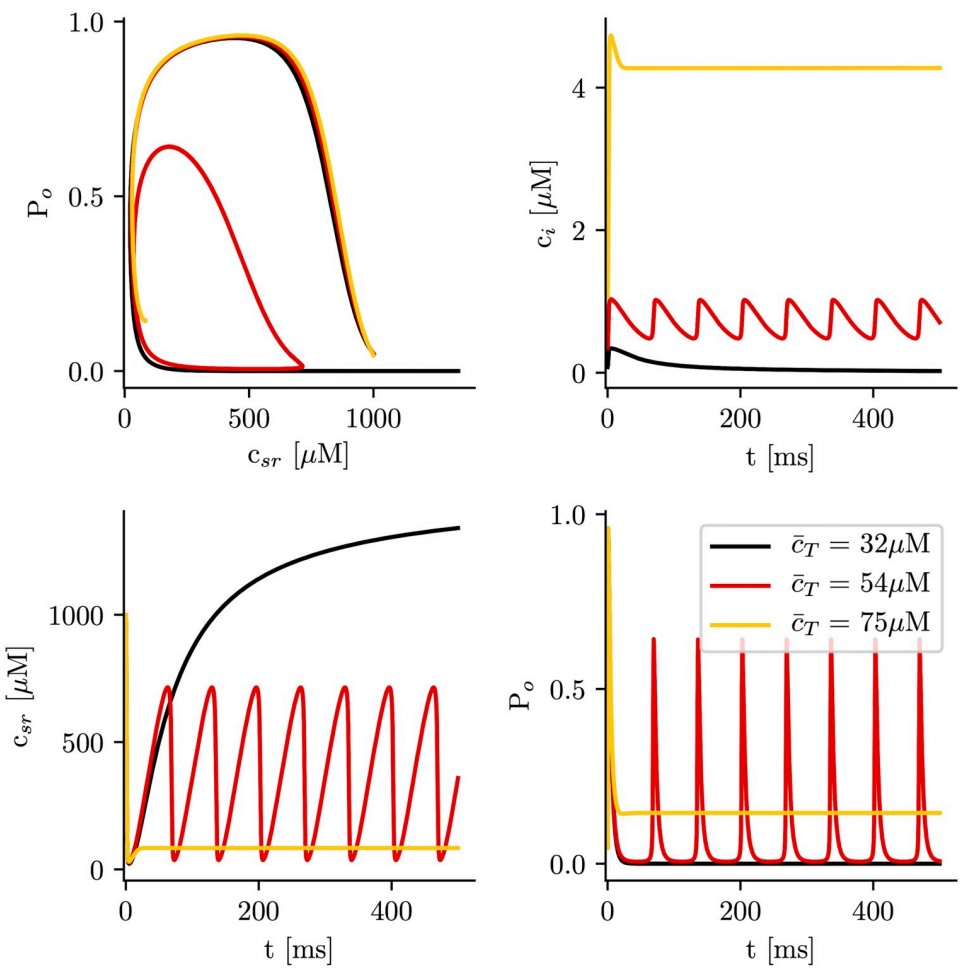

**Fig 14. Solutions as a function of total calcium concentration $\bar{c}_T$, with calsequestrin concentration set to zero when the fast dyadic approximation is used.** We obtain the same type of structure as expected. The system can be in a monostable state, which is excitatory (low load), in an oscillatory state (intermediate load), or in a bistable state (high loads), where it usually ends in a state of open RyR and depleted SR calcium concentration.

different slaving conditions may change the stability of the fixed points, that now are analyzed in the plane $(c_{sr}, P_o)$. Similarly to what we found in the previous analysis, calculating the stability, we find that the intermediate state is always unstable while the upper branch is stable above certain value of $\bar{c}_T$. Both states appear at around $\bar{c}_T \approx 54\mu M$, indicating the robustness of our analysis.

**Stochastic effects.** In real systems, the opening and closing of the RyR2 channels presents intrinsic stochastic dynamics. This is also the case in the full subcellular model. When considering average variables overall the cell, most of this stochasticity is averaged out. However, even if small, stochastic effects may play an important role [48] and, in particular, affect the properties of oscillations [49, 50]. We have thus studied these effects by including stochasticity in the minimal model (18) and (19), adding to the open probability ($P_o$) a small random fluctuation, such that:

$$P_o = \frac{c_d^2}{K_o^2 + c_d^2} + \sigma(U - 0.5) \tag{38}$$

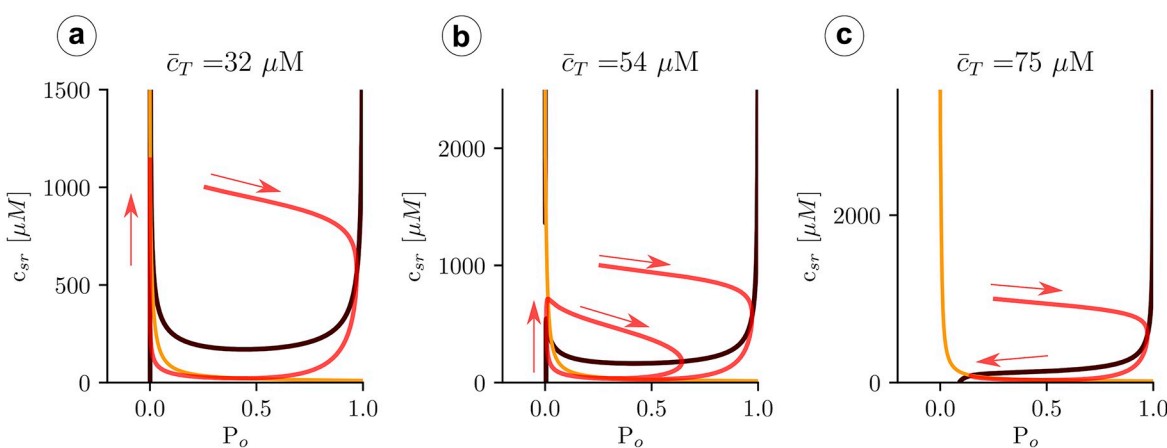

**Fig 15. Structure of the nullclines at different values of $\bar{c}_T$ indicated in the title of each panel.** The black line indicates the nullcline $\dot{P}_o = 0$, while the orange line corresponds to $\dot{c}_{sr} = 0$. The red curve is a trajectory with a direction indicated with the red arrows.

where $\sigma$ is the strength of the noise and $U$ is a random number between 0 and 1. The magnitude of $\sigma$ has been adjusted to be much smaller than $P_o$ when the system is in the open state or oscillates but large enough to be able to affect the dynamics close to the homoclinic bifurcation. In Fig 16a the calcium traces for both domain (cytosol and SR) are plotted. For high values of the load, the system oscillates in the same way as in the deterministic limit ($\bar{c}_T = 54\mu M$). In the deterministic model, the homoclinic bifurcation appears at $\bar{c}_T \simeq 50\mu M$, so it is reasonable to expect the same behavior for loads above this threshold. Below it, the oscillations in the stochastic system persist and the period increases as the load is decreased, as is typical for homoclinic bifurcations in the presence of noise [51]. For low values of $\bar{c}_T$ ($\bar{c}_T = 35\mu M$) the period has changed an order of magnitude ($\sim$ 1s) and now the calcium traces resemble closer those observed in the subcellular model, or in experiments [23]. The dependence of the period on the total load is shown in Fig 16b.

## Discussion and conclusion

Calcium oscillations play an important role in cardiac cells, from the regulation of growth in human cardiac progenitor cells [52], to the control of the pacemaker rhythm in both early embryonic heart cells [53, 54] and in sinoatrial nodal pacemaker cells (SANCs) [55–57]. Calcium oscillations have been observed under conditions of high cytosolic calcium concentration [23] or SR calcium overload. High levels of cytosolic calcium affect the opening probability of the RyR, which may result in oscillations or in a permanently open state [23, 58]. Calcium overload can be obtained, for instance, by inhibition of the Na$^+$-K$^+$ pump current $I_{NaK}$, that results in [Na$^+$]$_i$ overload. The consequent build-up of [Na$^+$]$_i$ reduces the effectiveness of the Na$^+$-Ca$^{2+}$ exchanger at removing calcium from the cell and intracellular calcium concentrations become elevated. A similar effect is observed in models of hypercalcemia [59]. The effect of elevated [Na$^+$]$_i$ has been studied in computational models, finding calcium oscillations [60], that, depending on the model, appear via a supercritical Hopf [24] or a homoclinic bifurcation [25, 26].

The instability of the diastolic resting state is thus well-known [61–63] and it plays an important role in the initiation of various cardiac arrhythmias [22]. However, the precise

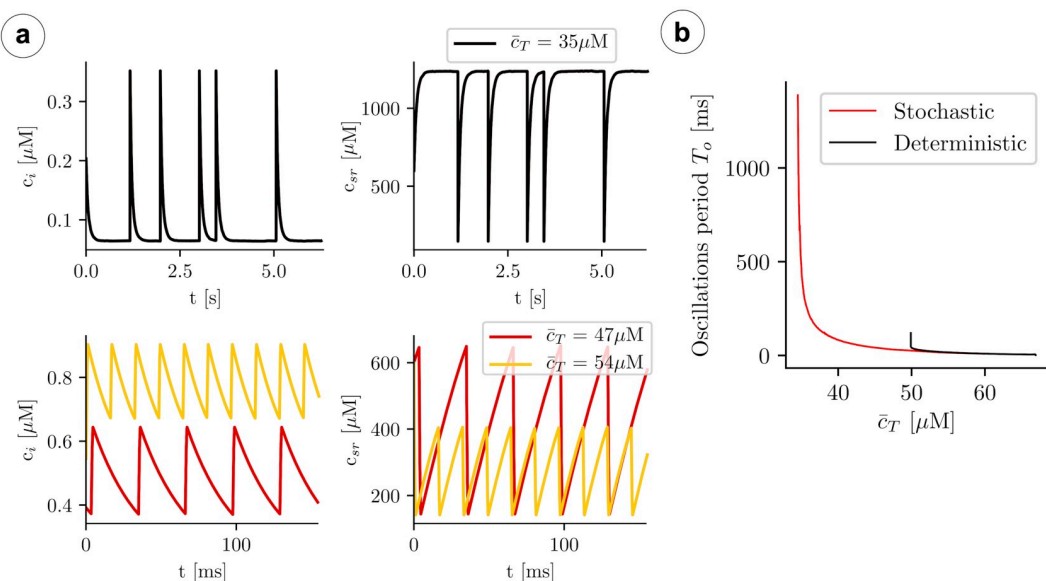

**Fig 16.** a) Traces of the cytosol and SR calcium concentrations for three different values of total calcium concentration $\bar{c}_T$. b) Oscillation periods as a function of $\bar{c}_T$. The strength of the noise is $\sigma = 2 \cdot 10^{-3}$.

mechanistic relationship between Ca and dangerous AP repolarization is still not well understood. One possibility that has been suggested in previous studies is that Ca cycling induces voltage deflections during the AP which can induce a substrate for wave break and reentry [64]. These voltage deflections are referred to as early-after-depolarizations (EADs), which can be caused by a variety of mechanisms, such as reactivation of the L-type Ca current during AP repolarization. An alternative possibility is that EADs may be caused by Ca oscillations that are regulated by Ca buffers such as CSQ. In this picture, calcium oscillations can perturb the AP via voltage sensitive currents such as the L-type Ca current and sodium-calcium exchange. This idea is consistent with several studies in the literature showing that mutations in CSQ cause abnormal Ca cycling rhythms that are directly linked to sudden cardiac death [65, 66]. Thus, a natural extension of our work would be to study the effect of calcium oscillations on the AP, for which we should extend our model to include detailed, and coupled, calcium and membrane dynamics.

We have shown, using a full subcellular model, that under global Ca overload, SR oscillations appear, leading finally to a state with permanently open RyR and depleted SR. In these simulations, oscillations give rise to periodic calcium waves, that propagate along the myocyte. To obtain a better understanding of the origin and parameter dependence of these transitions, we have studied them in a simplified model of the calcium dynamics. Despite not including all the physiological details (or because of that), minimal models are often useful to gain a better understanding of the origin of complex calcium rhythms [67], as oscillations [68]. We have thus analyzed the different transitions within a minimal model, that takes into account the RyR dynamics, as well as fluxes among different compartments. This allows us to explain the origin of oscillations using a nullcline analysis, as well as to give analytic expressions for the different transitions. A key conclusion is that all buffers can affect heavily the dynamics. The effect of buffers on oscillations has been studied previously [45], and experimentally a change in CSQ levels has been observed to alter the range of values of cytosolic calcium at which oscillations are observed [23]. Here we find that an increase in the levels of CSQ prevents altogether

the oscillations, obtaining a direct transition to an open state, that also occurs at larger values of total calcium content having, as mentioned, very important consequences for the study of EADs. We have shown that this picture is robust and independent on which is faster, whether the opening of the RyR2 or the diffusion of calcium near it. Although in this paper we have focused on the effect of CSQ, an interesting extension of this work would be to study in more detail the effect of changes in cytosolic buffer dynamics, allowing, for instance, for mobile buffers. This would require to modify the minimal model to account for difussion of the buffer to nearby calcium release units, and we defer it for future work.

The fact that oscillations appear both in a fully detailed stochastic model and in a simple deterministic model seems another proof of the robustness of the behavior. Our results with noise added to the minimal model seem to suggest that allowing the RyRs to behave stochastically could actually increase the parameter region where oscillations appear. This effect has been observed previously, where some amount of noise coming from the stochasticity of the ion channels may sustain oscillations by the process of coherence resonance [69, 70]. Thus, contrary to the results from the deterministic minimal model, where there is a sudden decrease in basal SR calcium load at the onset of oscillation, the addition of noise provides a gradual transition from the resting state to the oscillatory regime. This would agree with the results obtained by Stevens et al [23], where oscillations appeared gradually as cytosolic calcium content was increased, together with a continuous decrease in the level of basal SR. The addition of noise also gives a longer oscillation period, as observed in the subcellular model or in experiments. Even with this limitation, the analysis of the deterministic minimal model is still useful since it determines the minimal period of oscillation and the possibility of having periodic calcium waves. In a cell, when each calcium release unit presents or is very close to the onset of oscillations, periodic calcium waves appear. Under these circumstances, all units have a tendency to fire periodically, with noise making some units have a slightly early release. Diffusion of calcium to neighbors triggers the release of other units that were already close to open again. Waves are thus the coordinating mechanism of the oscillatory substrate of each unit.

Contrary to other scenarios studied for the onset of calcium waves [61], in our case, the rest state is not unstable, but rather it coexists with the oscillatory state, although, as discussed above, small fluctuations may favor the latter. Oscillations appear when the SR Ca flux through the RyRs becomes large, either because of a large luminal calcium load or to sensitization of the RyR due to an increase in diastolic calcium. These conditions can be met from a lack of CSQ, under an increase in global cell calcium content, or upon phosphorylation of the RyR, for instance. Under these circumstances, there is a strong release flux from the SR that results in SR depletion. As the SR is refilled, the SR release flux again increases, resulting in periodic oscillations. Although RyR inactivation is present in the subcellular model, in our simulations release refractoriness is due mostly to SR depletion, and its recovery is dictated by the slow refilling of the SR since this time scale is slower than the RyRs recovery from inactivation. Thus, the latter modulates the conditions of oscillations but does not determine them. We cannot discard, however, that under conditions of fast SR refilling, RyR refractoriness could not become the determining factor of the oscillations. This is reminiscent of the situation observed in calcium alternans, where both a slow refilling of the SR and RyR inactivation have been deemed as possible mechanisms for the onset of that rhythm [71, 72].

Finally, in this paper, we consider a cell which has achieved calcium balance where intrusion and extrusion match, and have neglected calcium fluxes across the cell membrane to focus on the internal calcium dynamics in order to decouple both processes. Under normal pacing, extracellular calcium fluxes typically represent about 10-20% of the total calcium fluxes, so it is not unreasonable to consider that the total calcium content remains constant once at a steady state. Under these conditions, cytosolic and SR calcium concentrations are not

independent but linked, and a clear control parameter is the total calcium concentration. Here we show that it uniquely determines the state of the system. Of course, in the presence of trans-membrane fluxes, calcium oscillations or waves, or a permanently open RyR, would result in an extrusion of calcium out of the cell and an eventual decrease in the total calcium load of the cell, that would transition back to the quiescent state (in the absence of external pacing). It seems interesting to study in the future the effect of oscillations and waves in the action potential, as well as a paced cell at different periods and the interaction with the pacing period. Observing how the time scale related to oscillations interacts with the time scale needed to extrude the calcium in the cytosol if the open (upper state) is reached may lead to new interesting phenomena.

## Supporting information

**S1 File. Supporting information file.** In this supplemental information file, we give a full description of the subcellular model, with a table of the different parameters used, as well as the effect of including inactivation in the RyR2. We also show how to obtain the minimal model from a reduction of a compartmental model of a CaRU.
(PDF)

**S1 Fig. Time traces of the different calcium concentrations for different values of total calcium concentration, and calsequestrin concentration set to zero (BSQ = 0), obtained simulating the three variable model (12)-(14).** After a transient, the system ends up in either a steady state which is excitatory at low levels of total calcium in the cell with observed low levels of calcium in the cytosol, in an oscillatory state with intermediate levels of total calcium in the cell, or in a state of high total levels of calcium in the cell with observed high cytosolic calcium levels.
(TIF)

**S2 Fig. Periodic calcium waves in the subcellular model with (a) and without (b) inactivation.** In the lower panel we show the fraction of RyRs in the different states shown in the schematics of Fig 1b. For the simulations in (b) we have set all the inactivation rates equal to zero.
(TIF)

## Acknowledgments

We want to thank I.R.Cantalapiedra, A.Peñaranda and L.Hove-Madsen from fruitful discussions.

## Author Contributions

**Conceptualization:** Miquel Marchena, Blas Echebarria, Yohannes Shiferaw, Enrique Alvarez-Lacalle.

**Data curation:** Miquel Marchena.

**Formal analysis:** Miquel Marchena, Blas Echebarria, Yohannes Shiferaw.

**Funding acquisition:** Blas Echebarria.

**Investigation:** Miquel Marchena, Blas Echebarria, Enrique Alvarez-Lacalle.

**Methodology:** Miquel Marchena, Yohannes Shiferaw, Enrique Alvarez-Lacalle.

**Software:** Miquel Marchena.

**Supervision:** Blas Echebarria, Enrique Alvarez-Lacalle.

**Validation:** Blas Echebarria, Yohannes Shiferaw, Enrique Alvarez-Lacalle.

**Visualization:** Blas Echebarria.

**Writing – original draft:** Miquel Marchena, Blas Echebarria, Enrique Alvarez-Lacalle.

**Writing – review & editing:** Miquel Marchena, Blas Echebarria, Yohannes Shiferaw, Enrique Alvarez-Lacalle.

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
