## [Decision Letter · Decision Letter 0]

15 Mar 2020

Dear Dr. Alvarez-Lacalle,

Thank you very much for submitting your manuscript "Buffering and total calcium levels determine the presence of oscillatory regimes in cardiac cells" for consideration at PLOS Computational Biology.

As with all papers reviewed by the journal, your manuscript was reviewed by members of the editorial board and by several independent reviewers. In light of the reviews (below this email), we would like to invite the resubmission of a significantly-revised version that takes into account the reviewers' comments.

We cannot make any decision about publication until we have seen the revised manuscript and your response to the reviewers' comments. Your revised manuscript is also likely to be sent to reviewers for further evaluation.

Sincerely,

Andrew D. McCulloch, Ph.D.

Associate Editor

PLOS Computational Biology

Daniel Beard

Deputy Editor

PLOS Computational Biology

Reviewer's Responses to Questions

**Comments to the Authors:**

Reviewer #1: The manuscript by Marchena et al. explores mechanisms underlying calcium oscillations in cardiac cells using mathematical models. They use a detailed model and a minimal model and show how buffering and total calcium levels affect calcium oscillations. The manuscript is well written and the findings are interesting. I have a few questions and suggestions.

*In the minimal model, Po only depends on cd (dyadic calcium). Doesn’t luminal Ca regulation of RyRs play a role in oscillations?

*Some buffers are fast, and others are slow. How do affinities, kinetics, and concentrations of the different buffers affect your results. Does mobility affect your results?

*Pg 3: “distributed over the cell along the z-lines with a Gaussian distribution in both transversal and longitudinal axes.” Can you elaborate this sentence? What are the mean and the standard deviation? Is there any experimental evidence?

*Pg7: “Waves are normally initiated at different sites each time” This implies that the cell size is also important (i.e. larger cells have more waves). On the other hand, the minimal model is a model of a single CaRU. Can you justify the use of the minimal model to analyze the full subcellular model?

*Fig 5: The oscillation period is too short for cardiac cells.

*Pg15: “we consider that the inactivated states of the RyRs do not play a role in the mechanisms that produce oscillations”. This assumption needs a reference.

*Pg16: “The parameters of these equations are taken from the literature where, except for those of the RyR, are well documented.” References are missing.

*It would be helpful if authors could write all the equations and parameters in the appendix so that readers do not have to read multiple papers to reproduce models and results.

*In my opinion, “We find, for instance” in the abstract is not appropriate. We want to know all the novel findings in the paper. Please enumerate all the key findings in the abstract.

*CSUN is located in Northridge (not in Los Angeles).

Reviewer #2: This manuscript is well written and in principle is a nicely designed sequence of model reduction for describing the dynamics underlying calcium instabilities in cardiac myocytes. However, I have considerable concern that the novelty of the findings are not sufficient to warrant publication in PLoS Comp Biol.

Major comments: The authors have provided a coherent analysis of 2 models, which are both reductions of their recently published model of subcellular calcium signaling in cardiac myocytes. In general I support their efforts to condense the complexity of the complete model to its essential dynamics. Previously this has been done to great effect for understanding EAD dynamics, and several times prior (sometimes in very similar fashion) for defining the stability of the calcium resting state in cardiomyocytes (see for example Tveito et al. 2012, Nivala et al. 2012). While I support the objectives in this way, I do not see the novelty in the current study to support publication at this time. In particular, the role of reduced SR calcium buffering in promoting instability of the diastolic resting state has been fully established experimentally and clinically as the second (and most severe) form of catecholaminergic polymorphic ventricular tachycardia (CPVT2). Similarly, the role of increased whole-cell calcium load in this form of instability has been well known for several decades, and specific critical SR calcium thresholds have been measured in a number of species (see many studies by Eisner’s group). In sum, my major comment here is that, while there has been great importance in prior studies that have used reduced models for describing the essential dynamics of instability e.g EADs, the real value of those analyses was permitted because there was an important disagreement between experimental observation, and the reigning concepts of the time. In the case of EADs this was the observation that APD could be prolonged tremendously without observing the transition to instability. I do not see any additional level of insight generated by the current study, and while it need not be groundbreaking, I think that some new biological insight is necessary to justify consideration for publication in PLoS Comp Biol.

Specific comments:

1. Was there some matter of simplicity that made it easier to base most of the nullcline analyses around the assumption of rapid gating kinetics i.e. dO_Ryr/dt = 0, rather than the assumption of rapid diffusion as is assumed to assess model robustness? I ask, because this assumption (quasi-equilibrium gating) is similar to that used to construct an Eigen value analysis for describing resting calcium stability (Tveito et al. 2012). Conversely, it has generally been assumed (for a range of model reduction exercises that remain faithful for CICR), that diffusion from the cleft is approximately an order of magnitude more rapid than the gating events themselves (see papers from Hinch in 2004).

2. With respect to the above point. The dynamics shown in Fig 5 are somewhat surprising from an intuitive perspective. This is because the SR [Ca] reached under stable conditions (C(bar)_T = 32 uM) is so much larger than for the other conditions (even though there should be no fundamental differences in the RyR properties, or in the relationship between free and total SR [Ca]). This makes me skeptical that the assumption of rapid gating is reasonable for describing the determinants of stability in this model. Can the authors explain to me what I am missing here?

**Have all data underlying the figures and results presented in the manuscript been provided?**

Reviewer #1: Yes

Reviewer #2: Yes

PLOS authors have the option to publish the peer review history of their article (what does this mean?). If published, this will include your full peer review and any attached files.

Reviewer #1: No

Reviewer #2: No
---

## [Decision Letter · Decision Letter 1]

15 Jun 2020

Dear Dr. Alvarez-Lacalle,

Thank you very much for submitting your manuscript "Buffering and total calcium levels determine the presence of oscillatory regimes in cardiac cells" for consideration at PLOS Computational Biology. As with all papers reviewed by the journal, your manuscript was reviewed by members of the editorial board and by several independent reviewers. The reviewers appreciated the attention to an important topic. Based on the reviews, we are likely to accept this manuscript for publication, providing that you modify the manuscript according to the one remaining review recommendation from reviewer 2.

Sincerely,

Andrew D. McCulloch, Ph.D.

Associate Editor

PLOS Computational Biology

Daniel Beard

Deputy Editor

PLOS Computational Biology

[LINK]

Reviewer's Responses to Questions

**Comments to the Authors:**

Reviewer #1: All of my comments have been addressed appropriately. I have no additional comments.

Reviewer #2: This revision has appropriately dealt with my comments and you have provided detailed responses and nicely reframed the paper.

I request one modification. My introductory comments (related to the value of reduced models in understanding the fundamental mechanisms of EADs) were simply to make a point about relating the core model behaviors to the missing information in the field. I suspect this may have driven you to include the section in the Discussion related to the involvement of these oscillatory behaviors on EADs. While I agree that the role of calcium cycling instability in EADs is clear, albeit in particular contexts, the involvement of the specific buffering-RyR gating regime that you have investigated here is not something that should be extended to that behavior without a full cell model incorporating detailed, and coupled, membrane dynamics. This would require a substantial investment that is beyond the scope of this paper. I recommend either removing the section relating this behavior to EADs, or to reduce its emphasis and make it very clear that such an involvement is highly speculative without verification in a comprehensive model of myocyte calcium and electrophysiology.

**Have all data underlying the figures and results presented in the manuscript been provided?**

Reviewer #1: Yes

Reviewer #2: No: There is no note of an available code repository for this model or simulations

PLOS authors have the option to publish the peer review history of their article (what does this mean?). If published, this will include your full peer review and any attached files.

Reviewer #1: No

Reviewer #2: Yes: Andrew G Edwards
---

## [Editor Report · Decision Letter 2]

7 Jul 2020

Dear Dr. Alvarez-Lacalle,

We are pleased to inform you that your manuscript 'Buffering and total calcium levels determine the presence of oscillatory regimes in cardiac cells' has been provisionally accepted for publication in PLOS Computational Biology.

Best regards,

Andrew D. McCulloch, Ph.D.

Associate Editor

PLOS Computational Biology

Daniel Beard

Deputy Editor

PLOS Computational Biology

---

## [Editor Report · Acceptance letter]

17 Sep 2020

PCOMPBIOL-D-20-00218R2 

Buffering and total calcium levels determine the presence of oscillatory regimes in cardiac cells

Dear Dr Alvarez-Lacalle,

I am pleased to inform you that your manuscript has been formally accepted for publication in PLOS Computational Biology. Your manuscript is now with our production department and you will be notified of the publication date in due course.

With kind regards,

Matt Lyles
